# Tissue-Specific ^1^H-NMR Metabolomic Profiling in Mice with Adenine-Induced Chronic Kidney Disease

**DOI:** 10.3390/metabo11010045

**Published:** 2021-01-10

**Authors:** Ram B. Khattri, Trace Thome, Terence E. Ryan

**Affiliations:** 1Department of Applied Physiology and Kinesiology, University of Florida, Gainesville, FL 32611, USA; rbk11@ufl.edu (R.B.K.); trthome@ufl.edu (T.T.); 2Center for Exercise Science, University of Florida, Gainesville, FL 32611, USA; 3Myology Institute, University of Florida, Gainesville, FL 32611, USA

**Keywords:** CKD, uremia, metabolism, muscle, cardiac, liver, catabolism

## Abstract

Chronic kidney disease (CKD) results in the impaired filtration of metabolites, which may be toxic or harmful to organs/tissues. The objective of this study was to perform unbiased ^1^H nuclear magnetic resonance (NMR)-based metabolomics profiling of tissues from mice with CKD. Five-month-old male C57BL6J mice were placed on either a casein control diet or adenine-supplemented diet to induce CKD for 24 weeks. CKD was confirmed by significant increases in blood urea nitrogen (24.1 ± 7.7 vs. 105.3 ± 18.3 mg/dL, *p* < 0.0001) in adenine-fed mice. Following this chronic adenine diet, the kidney, heart, liver, and quadriceps muscles were rapidly dissected; snap-frozen in liquid nitrogen; and the metabolites were extracted. Metabolomic profiling coupled with multivariate analyses confirm clear separation in both aqueous and organic phases between control and CKD mice. Severe energetic stress and apparent impaired mitochondrial metabolism were observed in CKD kidneys evidenced by the depletion of ATP and NAD^+^, along with significant alterations in tricarboxylic acid (TCA) cycle intermediates. Altered amino acid metabolism was observed in all tissues, although significant differences in specific amino acids varied across tissue types. Taken together, this study provides a metabolomics fingerprint of multiple tissues from mice with and without severe CKD induced by chronic adenine feeding.

## 1. Introduction

Chronic kidney disease (CKD) impacts an estimated 8–16% of the world’s population [1] and is a significant risk factor for mortality [2]. Unfortunately, CKD has no cure, with the available treatment options being limited to chronic dialysis or kidney transplant. In regards to the latter option, available transplant numbers are far outpaced by the increasing prevalence of CKD. Proper kidney function is required for the maintenance of systemic homeostasis, including the regulation of electrolytes, fluid levels, blood pressure, acid–base balance, filtration of metabolic waste, and production of hormones, among many others. The impaired glomerular filtration that accompanies CKD results in the retention and accumulation of metabolic by-products, which are either produced endogenously or ingested. Many of these metabolic by-products have been described as uremic toxins because of their harmful effects on tissues/organs [3,4,5,6,7,8,9]. Some of these metabolites have been reported to disrupt cellular metabolic pathways [8,10,11,12,13,14,15,16,17], which may be especially harmful to organs with high levels of energy (ATP) demand, such as the heart, kidney, liver, and skeletal muscles, which, together, account for ~60–65% of the daily energy expenditure.

Advances in technology have given rise to the relatively new field of Metabolomics, which is focused on the study of metabolites in biological systems [18]. Using mass spectrometry and NMR, many different metabolites can be accurately identified and quantified with relatively small tissue requirements [19], which can provide unprecedented biochemical information about the function of tissues/organisms. These analyses could help identify novel therapeutic targets to counteract the negative impacts of uremia/CKD or discover new biomarkers aiding in the diagnosis or prognosis of CKD.

The objective of this study was to perform metabolomic analyses of the heart, liver, kidney, and skeletal muscles obtained from mice with and without severe CKD. To accomplish this objective, we extracted tissues from mice with CKD induced by the long-term (24 weeks) consumption of an adenine-supplemented diet, as well as their control diet-fed counterparts with normal kidney functions. Metabolites were extracted from tissues, and ^1^H nuclear magnetic resonance (NMR) was performed and coupled with multivariate statistical analysis.

## 2. Results

### 2.1. Long-Term Adenine Feeding Induces Severe CKD

Adenine diet supplementation is an established model to study CKD in rodents [20,21,22,23,24,25] that causes tubulointerstitial nephropathy, resulting from an increased production of 2,8-dihydroxyadenine, which crystallizes in the renal tubules, causing tubular damage and fibrosis. In this study, mature five-month-old male C57BL/6J mice were provided either a casein control or adenine-supplemented diet for 24 weeks (approximately six months), as shown in Figure 1. CKD was confirmed by a substantial increase in blood urea nitrogen (BUN) in CKD mice compared to controls (24.1 ± 7.7 mg/dL vs. 105.3 ± 18.3 mg/dL, *p* < 0.0001, *n* = 7/group). Furthermore, CKD mice displayed much lower body weights compared to casein controls (32.86 ± 1.21 g vs. 23.57 ± 1.27 g, *p* < 0.0001, *n* = 7/group), consistent with the reported catabolic state cause by CKD [26,27,28,29,30]. Moreover, the clinical disease stage was quite severe, evidenced by the spontaneous death of three CKD mice, which occurred within two weeks prior to sacrifice. Tissues from the three mice that were found dead were not used in the metabolomics analyses described below.

### 2.2. H NMR Spectroscopic Analysis of Tissue Extracts

FOLCH extraction was used to obtain both aqueous and organic phase metabolites from the kidney, heart, liver, and skeletal muscle (quadriceps) of control and CKD mice. Metabolite extracts were analyzed using ^1^H NMR. Representative ^1^H NMR spectra for both control and CKD tissues are shown in Figure 2 (aqueous phase) and Appendix A. The spectra were assigned with the help of Chenomx Suite 8.6 NMR software (Chenomx, Inc., Edmonton, AB, Canada), the previous literature, the biological magnetic resonance bank (BMRB) [31], and a set of 2D spectra (Appendix A). The aqueous phase samples contained a number of metabolites, including amino acids, tricarboxylic acid (TCA) cycle intermediates, glycolytic metabolites, pyruvate, creatine, creatinine, and trimethyl-N-oxide, as well as energetic metabolites such as NAD^+^, AMP/ATP, and inosine/adenosine. The organic phase samples contained a mixture of fatty acids (saturated and unsaturated), cholesterol and cholesterol esters, phosphatidylcholine, choline, sphingomyelin, phospholipids, glycerophospholipids, and triglyceride. A complete list of identified metabolites and their respective quantifications can be found in Appendix A.

### 2.3. Multivariate Analysis of ^1^H NMR Spectroscopic Data

Intrinsic variations within a tissue sample in between the control and CKD groups were determined through principal components analysis (PCA), partial least squares discriminant analysis (PLS-DA), and orthogonal projections to latent structures discriminant analysis (OPLS-DA) using false discovery rate (FDR) corrected ^1^H NMR data. Results from the PCA analysis (Figure 3) demonstrate clear clustering among control and CKD groups. PCA components 1 and 2 account for 60–75% of the total variance between the control and CKD groups for both aqueous and organic phase samples from all tissues.

The PCA method, since being unsupervised, explores the magnitude of variation present in-between the groups [32]. Next, we also performed supervised PLS-DA (Appendix A) and OPLS-DA analyses (Figure 4) to maximize the separation between the control and CKD. A combination of PCA with PLS-DA and OPLS-DA analyses provided a more comprehensive metabolomics fingerprint that not only tried to explore the variations among the groups but, also, helped to discern those variables driving the maximal separation. The PLS-DA analysis demonstrated a strong separation between the control and CKD for both the aqueous phase (Appendix A) and organic phase (Appendix A) samples. The model predictability (Q^2^ value) was above 0.4 for all samples, indicating excellent goodness of fit within the model. The metabolites/compounds that were responsible for driving the separation for all tissue samples (obtained from the variable importance in projection (VIP) scores of the PLS-DA analysis) are shown in Appendix A.

A clear differentiation between the control and CKD metabolomic fingerprints can be observed in the OPLS-DA score plots, especially for water-soluble metabolites (Figure 4A–D). The lipid-soluble metabolites also showed good separation between the control and CKD groups, particularly in the heart and liver (Figure 4F,G, respectively). However, the lipid-soluble metabolites had a slight overlap in both the kidney and quadriceps muscle (Figure 4E,H, respectively). Furthermore, the Q^2^ value demonstrated a high goodness of fit, with values of 0.82, 0.61, 0.72, and 0.69 for the aqueous phase samples of the kidney, heart, liver, and quadriceps, respectively. Similarly, for organic phase samples, the Q^2^ values found were 0.57, 0.3, 0.72, and 0.28 for the kidney, heart, liver, and quadriceps, respectively.

### 2.4. CKD Results in Tissue-Specific Metabolomic Fingerprints, Indicating Severe Energetic Stress in Kidneys

CKD and the associated retention of uremic toxins clearly had a strong impact on the metabolite profiles. Heatmaps for log_2_-transformed fold changes in metabolites in the CKD samples are shown in Figure 5A (aqueous phase) and Figure 5B (organic phase). As described above, a complete list and quantification of these metabolites can be found in Appendix A.

Unsurprisingly, the kidney generally displayed larger magnitude changes in metabolite abundance compared with the other tissues. A remarkable feature of the kidney metabolome appears to indicate a clear energetic stress, consistent with a previous report documenting a failure of the kidney to sense energy depletion [33]. For example, kidneys from adenine-fed mice displayed significant reductions in ATP (0.34-fold control) and NAD^+^ (0.12-fold control) (Figure 6). The kidney also exhibited changes in a number of TCA cycle intermediates, including decreased levels of succinate, fumarate, and glutamate and a six-fold increase in citrate (Figure 6). These findings implicate the disruption of dehydrogenase enzyme activity within the mitochondrial matrix. Interestingly, glutamate was the only TCA cycle metabolite to be significantly decreased in all tissues. In contrast to the kidney and skeletal muscle, fumarate was significantly increased in the livers of CKD mice (Figure 6).

### 2.5. Altered Amino Acid Metabolism in CKD

Another notable and rather consistent metabolic phenotype observed in CKD tissues was altered amino acid profiles (Figure 7). CKD is known to be a catabolic disease, resulting in protein-energy wasting and whole-body and muscle protein loss [26,27,34,35,36]. Branched chain amino acids (BCAA) isoleucine, leucine, and valine were significantly decreased in the kidneys and hearts of CKD mice, with valine also being decreased in the skeletal muscle. The kidneys again displayed the largest changes, with significant decreases in several other amino acids, including tyrosine, aspartate, glycine, alanine, histidine, and glutamine. The skeletal muscle also exhibited decreased tyrosine, alanine, and histidine levels in CKD mice. The liver exhibited the lowest disruption in amino acid profiles of the tissues analyzed, with significant decreases only observed in lysine, aspartate, and tyrosine, with the BCAA levels being relatively normal.

### 2.6. Lipidomic Changes Caused by CKD

Dyslipidemia is commonly reported in CKD patients [37] and characterized by elevated blood levels of triglycerides and cholesterol. Although this study did not evaluate the blood, an analysis of organic phase NMR spectra from the kidney, liver, heart, and quadriceps muscles also indicated significant alterations of the lipid species caused by CKD (Figure 8). In contrast to the blood, significant decreases in triglycerides were found in the kidney, liver, and heart samples from CKD mice (Figure 8A–C). Cholesterol, a major component of cellular or organelle membranes, appeared to be decreased in CKD kidneys, hearts, and, potentially, livers; however, the cholesterol levels were unchanged in the skeletal muscles (Figure 8D–F). Saturated and unsaturated fatty acid species we only decreased in the kidney and liver in CKD mice (Figure 8G–I). Phospholipids and glycerophospholipid compounds were markedly decreased in the skeletal muscle and liver (Figure 8J–L). Notably, the CKD kidney displayed significant decreases in almost every (14 out of 17) lipid compound that was measurable by NMR.

## 3. Discussion

This study utilized ^1^H NMR-based metabolomics to evaluate changes in the metabolites in mice with long-term adenine-induced CKD. Fully matured five-month-old male mice were provided either a casein control or adenine-supplemented diet for approximately six months. Adenine-fed mice exhibited symptoms of severe or end-stage CKD, including high BUN levels and a ~30% loss in body weight. A metabolomic analysis of the kidney, heart, liver, and quadriceps skeletal muscle were found to be clearly distinguishable between control and CKD mice using multivariate analyses (Figure 3 and Figure 4). These observations demonstrated the profound and, in some instances, tissue-specific metabolic impacts of chronic renal insufficiency and the ensuing uremic condition.

Metabolomic profiling of adult mice chronically (six months) fed an adenine-supplemented diet identified a substantial energetic stress within the kidneys. Specifically, kidney metabolomics in CKD mice demonstrated a significant decrease in ATP and NAD^+^ levels, as well as reductions in other TCA cycle intermediates, including fumarate, succinate, and glutamate (Figure 6). A roughly six-fold increase in citrate and significant increase in glucose was also observed, consistent with impaired TCA cycle flux [38]. These findings of energetic stress within the kidney are consistent for previous studies. Kang et al. reported in both humans and a mouse model of CKD that a lower expression of enzymes and regulators of fatty acid oxidation is the key driver of kidney disease and that increasing fatty acid oxidation could protect against pathological changes within the kidney [39]. Similarly, Kikuchi et al. reported a substantial increase in the kidney AMP:ATP ratio within the kidney of mice subjected to a 5/6th nephrectomy surgical model of CKD [33]. As mitochondria are the principal site of ATP production in the kidney, the energy depletion and accumulation of NAD^+^ coupled with the alterations in the TCA cycle intermediates suggest the presence of a severe metabolic mitochondrial pathology within the CKD kidney. Taken together, future studies may seek to develop a novel treatment to improve kidney bioenergetics as a means of increasing kidney function and reducing pathology.

CKD is well-known as a catabolic disease in which patients display reduced body weight and muscle atrophy/wasting that compromises their strength and physical functions [26,34,35,40]. Unfortunately for patients, this catabolic state has been shown to increase with hemodialysis [26], one of the primary treatment options for CKD. While much has been learned about muscle catabolism, little is known about protein catabolism in other organs/tissues in the context of CKD/uremia. Amino acid profiling of the tissues in this study confirmed aberrant amino acid catabolism in CKD mice (Figure 7). The BCAA (isoleucine, leucine, and valine) were all significantly decreased in hearts and kidneys, whereas valine was the only BCAA decreased in the skeletal muscle. Notably, liver BCAAs were unchanged in CKD. Decreased tissue amino acids were likely indicative of increased protein catabolism and, presumably, resulted in increased serum/blood amino acid levels. This would align with a recent study that reported a significant inverse association between the serum amino acid levels and incident CKD in a large clinical population [41].

An analysis of the extracted organic phase, which contained primarily lipid compounds, also uncovered a substantial change with CKD, particularly in the kidney and liver (Figure 8). Interestingly, the kidney from CKD mice displayed lower abundances of triglycerides, cholesterols, and saturate and unsaturated fatty acid species, as well as phospholipids. These results indicate the impacts of lipid homeostasis, including impaired oxidation, which has been previously shown in human plasma from CKD patients [42,43], as well as possible impairments in lipid biosynthesis that may be rampant within the failing kidney. The progressive nature of CKD may also impart time- or disease severity-dependent changes in the lipid homeostasis caused by the progressive decline in renal function and the likely adaptive mechanisms within various organs. Much like aqueous phase metabolites, lipid profiles in CKD were unique by tissue type. For example, only phospholipid species were altered in CKD skeletal muscle, and the heart showed differences primarily in cholesterols and triglycerides. The differential abundance of lipids in this study belong to several different biosynthetic and metabolic pathways. Although we did not explore alterations in the expression of genes or proteins (enzymes) in these metabolic pathways, there is previous evidence suggesting that these could be altered by CKD [42,43,44].

There are some noteworthy limitations of the present study. First, this study was performed only in male mice, although CKD is quite prevalent in female patients. Future metabolomics analyses should be performed in female mice with severe CKD and may also consider the potential impact of sex hormones, such as estrogens, which have renoprotective effects [21,45]. There are several approaches to metabolomics analyses involving primarily mass spectrometry or NMR-based technologies. Mass spectrometry approaches offer lower limits of detection but have lower reproducibility compared with NMR. Thus, the metabolomic fingerprints developed in this study are not able to detect the many metabolites present in the tissues at lower concentrations. Metabolites were extracted from flash-frozen tissue specimens that contained multiple cell types. For example, the kidney has a very distinct structure with many cell types that perform vastly different functions within the tissue [46]. Therefore, the metabolite changes presented herein cannot be directly attributed solely to a single cell type/population within the respective tissues. Finally, there are numerous preclinical models of CKD used in the field, each with numerous benefits and limitations discussed previously [47,48,49]. None of the available preclinical CKD models, including the adenine model employed herein, perfectly match the human disease etiology and pathology. Despite these limitations, obtaining human patient tissue samples from these vital organs is extremely difficult, and there are certain aspects of the adenine model that align with the clinical phenotype, including uremia, reduced glomerular filtration, and renal fibrosis and interstitial inflammation [25].

In summary, this study performed 1H NMR metabolomics analyses of the kidney, liver, heart, and quadriceps muscle of mice with CKD. CKD was induced by the feeding of adenine chronically for six months. To our knowledge, this is the longest duration of adenine-induced CKD in mice ever reported. CKD results in tissue-specific metabolomic profiles in which the largest changes are observed in the kidney itself. Extreme energetic stress was a key feature of the CKD kidney, whereas most tissues exhibited signs of aberrant amino acid metabolism, primarily indicating a highly catabolic state. These metabolic alterations provide a foundation in which the exploration of metabolic therapies can be developed, with the goal of improving CKD outcomes. Future studies employing ^1^H-NMR and mass spectrometry-based metabolomics analyses are warranted to continue increasing our understanding of the broad and tissue-specific metabolic impacts of impaired renal function. These studies should extend beyond preclinical models to the human patient. Finally, mechanistic studies investigating the physiological impact and potential therapeutic opportunity of altering metabolisms are needed in CKD.

## 4. Materials and Methods

### 4.1. Chemicals

Sodium monobasic phosphate, sodium dibasic phosphate, sodium azide (NaN_3_), pyrazine, and ethylene diamine tetra acetic acid (EDTA) were purchased from Sigma Aldrich (St Louis, MO, USA). Cambridge Isotope Laboratories (Andover, MA, USA) supplied the deuterated solvents, such as chloroform (CDCl_3_) and deuterium oxide (D_2_O). D_6_-4,4-dimethyl-4-silapentane-1-sulfonic acid (D_6_-DSS) was purchased from FUJIFILM Wako (Richmond, VA, USA).

### 4.2. Animals

Male C57BL/6J mice (Stock #000664) were obtained from Jackson Laboratory at 5 months of age (*n* = 17 mice total). All rodents were housed in a temperature (22 °C) and light-controlled (12-hour light/12-hour dark) room and maintained on standard chow diet until the induction of CKD with free access to food and water (3–5 animals per cage). All animal experiments adhered to the Guide for the Care and Use of Laboratory Animals from the institute for Laboratory Animal Research, National Research Council, Washington D.C., National Academy Press, 1996, and any updates. All procedures were approved by the Institutional Animal Care and Use Committee of the University of Florida under protocol 201810484.

### 4.3. Induction of Chronic Kidney Disease (CKD)

To induce CKD, we utilized an established adenine diet model [20,23,24,25]. Following acclimation to the vivarium, mice were all provided a casein-based chow for seven days, followed by random assignment to either the adenine-supplemented diet (0.2%) or casein control diet for the next seven days. After one week on 0.2% adenine, the CKD group was switched to a 0.15% adenine diet for 24 weeks. CKD mice were then placed back on the casein control diet for two weeks prior to euthanasia and tissue harvest, allowing for a washout period of adenine-based chow. Control mice received the casein diet for the duration of the study.

### 4.4. Assessment of Blood Urea Nitrogen (BUN)

BUN was assessed in the plasma of control and CKD mice. Prior to sacrifice, blood was collected in a heparin-coated capillary tube from a ~1-mm tail snip. Following collection, blood was spun down at 4000 rpm for ten minutes at 4 °C, and plasma was collected. BUN was assessed using a commercial kit (Arbor Assays, Ann Arbor, Michigan, USA; Cat # K024) according to the manufacturer’s instructions.

### 4.5. Tissue Collection

While under isoflurane anesthesia, tissues were rapidly dissected and snap-frozen in liquid nitrogen and stored at −80 °C until metabolite extraction. The following tissues were used in this study: kidney, liver, heart (left ventricle), and skeletal muscle (quadriceps). Euthanasia was carried out by thoracotomy followed by cervical dislocation.

### 4.6. Extraction of Metabolites

A modified form of the FOLCH extraction [50] protocol was used to extract metabolites from the tissues. Wet weights of all tissue samples were recorded prior to extraction. Tissue samples were immediately homogenized to prevent any possible enzymatic action using 1 mL of ice-cold methanol in a PowerLyzer 24 Homogenizer (QIAGEN Group, Hilden, Germany). The mixture was centrifuged using 13,200 rpm at 4 °C for 30 min, and the resulting supernatant was transferred to a new glass vial consisting of 3 mL of ice-cold chloroform:methanol (2:1, *v*/*v*) mixture. The homogenate was vortexed and left in an ice bath for 15 min to allow for phase separation. Next, 1 mL of 0.9% of saline was added and vortexed for a couple of minutes, followed by a second incubation in an ice bath for 30–45 min for complete phase separation. The upper aqueous layer was transferred to a new falcon tube. To the remaining organic phase sample, 1 mL of 0.9% of saline was added again, followed by vigorous mixing and letting it stand in an ice bath (15 min) for a second phase separation. This second aqueous phase was combined with the first. The resulting aqueous and organic layers were dried separately. The aqueous layer was dried overnight with a Labconco freezer dryer (Labconco Corporation, Kansas city, MO, USA), and the organic layer was dried via inert nitrogen gas. These two dried powders (aqueous and organic phases) were stored at −80 °C until performing NMR experiments.

### 4.7. NMR Sample Preparation and Spectra Acquisition

Proton spectra were collected for both sets of samples (aqueous and organic phases) using a Bruker Avance Neo 14.1 T (600 MHz) (Bruker BioSpin Corporation, Billerica, MA, USA) equipped with a 1.7-mm TCl CryoProbe. Aqueous phase samples were dissolved in 45 µL of 50-mM phosphate buffer (at pH 7.2), along with 5 µL of Chenomx standard (Chenomx, Inc., Edmonton, AB, Canada). The Chenomx standard had 5 mM of D_6_-DSS. The buffer mixture was in 100% deuterated environment and, also, supplemented with 2 mM of EDTA and 0.2% of NaN_3_. Deuterated chloroform (80 µL) consisting of 10 mM of pyrazine (as the internal NMR standard) was used to resuspend the organic phase samples. One-dimensional spectra were acquired using the 1D nuclear Overhauser effect spectroscopy (NOESY) pulse sequence [51] using the parameters reported previously [32,52,53]: 1-s recycle delay(d1), 4-s acquisition time (acq), 128 scans (nt), 12-ppm spectral width (sw), 100-ms mixing time, ^1^H 90° pulse width (pw), 25 °C temperature, and 8-s dummy scans. Water pre-saturation power was applied during the recycle delays for the aqueous phase samples.

### 4.8. Data Processing and Analysis

Processing of spectra was performed in MestReNova 14.1.2-25024 software (Mestrelab Research, S.L., Santago de Compostela, Spain). Line broadening of 0.22 Hz was applied, along with 64k data point zero filling prior to Fourier transformation. The spectra were further phase-corrected, followed by Splines/Whittaker Smoother baseline correction. Water-soluble metabolite spectra were referenced and normalized with a D_6_-DSS (internal standard) peak at 0.00 ppm. Chloroform-soluble metabolite/compound spectra were referenced with respect to (w.r.t.) CDCl_3_ peak (7.26 ppm) followed by a normalization w.r.t. pyrazine (internal standard) peak at 8.61 ppm. Integrated peak areas were extracted from the normalized spectra and utilized for quantitative purpose after wet weight normalization. Furthermore, quantification of some metabolites with overlapping peaks was performed using Chenomx Suite 8.6 NMR software (Chenomx, Inc., Edmonton, AB, Canada). Metabolites were assigned and verified with the aid of the literature [54,55] and biological magnetic resonance bank (BMRB) [31], as well as 1D and 2D spectra collected for a particular sample. Two-dimensional spectra (correlation spectroscopy [56], total correlation spectroscopy [57], heteronuclear single-quantum correlation spectroscopy [58], and heteronuclear multiple-bond correlation spectroscopy [57]) were collected utilizing the standard Bruker library (Appendix A).

### 4.9. Statistical Analysis

Online web-based Metaboanalyst4.0 (https://www.metaboanalyst.ca/) software was used to perform principal component analysis (PCA) and partial-least square discriminant analysis (PLS-DA), and orthogonal partial least-square analysis (OPLS-DA) was also performed using Metaboanalyst4.0. We extracted peak integration areas from the preprocessed NMR spectra and normalized the peak areas to the tissue wet weight. These normalized peak areas were input to Metaboanalyst4.0 for further analysis. Interquartile range (IGR) filtering was applied to exclude any peaks that did not represent biological metabolites (i.e., NMR noise). Data variability was corrected using probability quotient normalization, followed by pareto-scaling on the false discovery rate (FDR) correct data. Q^2^ test was applied to test the validity of the PLS-DA and OPLS-DA methods. Metabolites/compounds with variable importance in projection (VIP) scores ≥ 1 were considered to significantly drive the separation between control and CKD groups. Between-group comparisons were performed using a two-tailed, unpaired Student’s *t*-test, with *p* < 0.05 considered statistically significant.

GraphPad Prism (version 9.0.0 (121), GraphPad Software, San Diego, CA, USA, www.graphpad.com) was utilized to create Box and Whisker plots. In all cases, data were presented as mean ± standard deviation.

## Figures and Tables

**Figure 1 metabolites-11-00045-f001:**
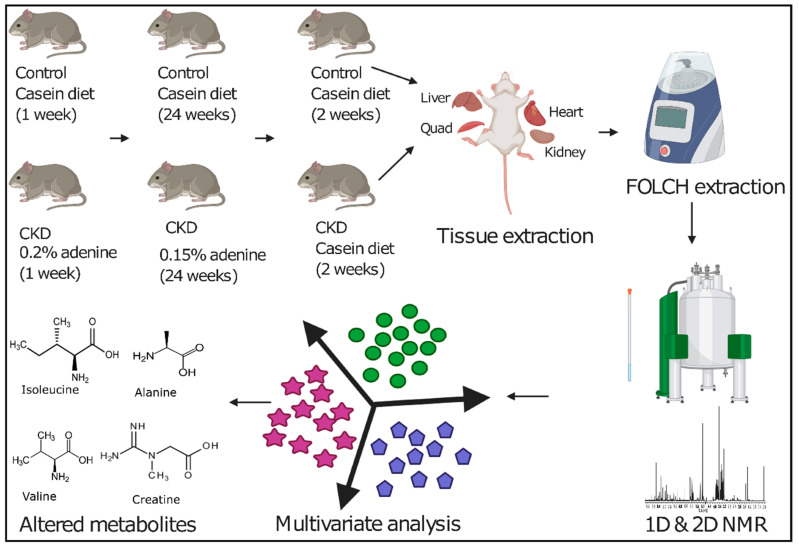
Graphical representation of the overall workflow employed in this study that includes adenine and casein diet supplementation in mature C57BL/6J male mice, tissue collection, FOLCH extraction, ^1^H NMR spectroscopy, multivariate analysis, and altered metabolite identification. CKD: chronic kidney disease.

**Figure 2 metabolites-11-00045-f002:**
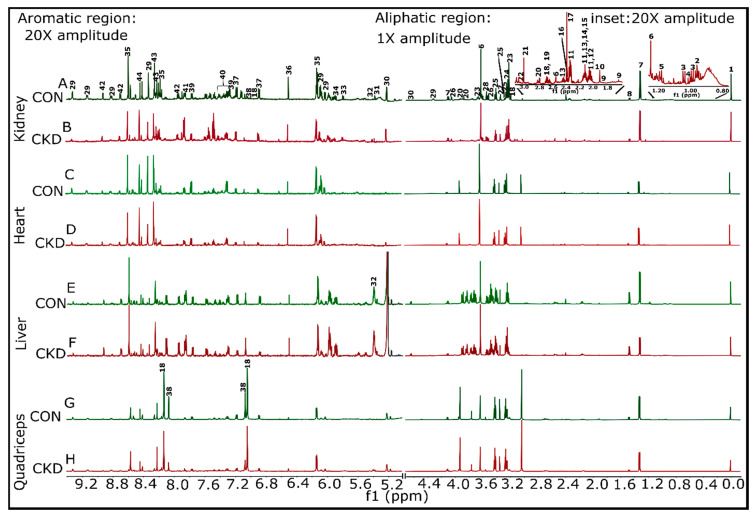
Representative ^1^H NMR spectra for the aqueous phase samples from both control and CKD groups were shown for kidney, heart, liver, and quadriceps muscle. Samples having almost equal wet weigh for both control and CKD groups within a particular tissue sample were selected so that the comparisons become legitimate. (**A**,**B**) Kidney control#5 (12.3 mg) and CKD#5 (12.5 mg), respectively. (**C**,**D**) Heart control#5 (6.8 mg) and CKD#5 (6.8 mg), respectively. (**E**,**F**) Liver control#4 (13.2 mg) and CKD#4 (12.0 mg), respectively. (**G**,**H**) Quadriceps control#6 (14.7 mg) and CKD#6 (15.2 mg), respectively. (A) 1 represents the 4,4-dimethyl-4-silapentane-1-sulfonic acid (DSS peak), 2 is leucine, 3 is valine, 4 is isoleucine, 5 is ethanol, 6 is Ca^2+^-complexed ethylene diamine tetra acetic acid (EDTA), 7 is lactate, 8 is alanine, 9 is lysine, 10 is acetate, 11 is glutamate, 12 is proline, 13 is glutamine, 14 is methionine, 15 is O-acetyl choline, 16 is pyruvate, 17 is succinate, 18 is anserine, 19 is citrate, 20 is asparate, 21 is creatine, 22 is creatinine, 23 is Sn-glycero-3-phosphocholine, 24 is trimethyl-N-oxide, 25 is taurine, 26 is Myo-inositol, 27 is methanol, 28 is glycine, 29 is NAD^+^, 30 is glucose, 31 is allantoin, 32 is glucose-1-phosphate, 33 is uracil, 34 is uridine triphosphate, 35 is AMP + ATP, 36 is fumarate, 37 is tyrosine, 38 is histidine, 39 is tryptophan, 40 is phenylalanine, 41 is benzoate, 42 is nicotinurate, 43 is inosine/adenosine, and 44 is formate. Proton spectra were collected for both sets of samples (aqueous and organic phases) using a Bruker Avance Neo 14.1 T [600 MHz] (Bruker BioSpin Corporation, Billerica, MA, USA) equipped with a 1.7-mm TCl CryoProbe.

**Figure 3 metabolites-11-00045-f003:**
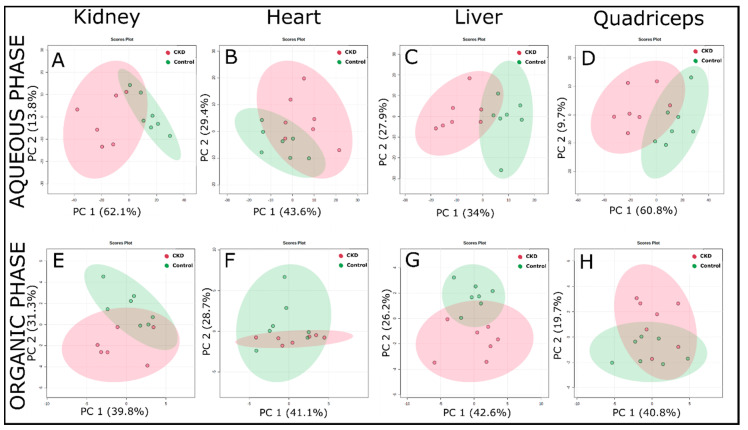
Principal component analysis (PCA) analysis of the ^1^H-NMR spectra of the extracted tissue samples (using a targeted profiling approach) from control and adenine-induced CKD groups. (**A**–**D**) PCA score plots from the aqueous phase samples for the kidney, heart, liver, and quadriceps, respectively. (**E**–**H**) PCA score plots from the organic phase samples in the same order as mentioned above (*n* = 7/group).

**Figure 4 metabolites-11-00045-f004:**
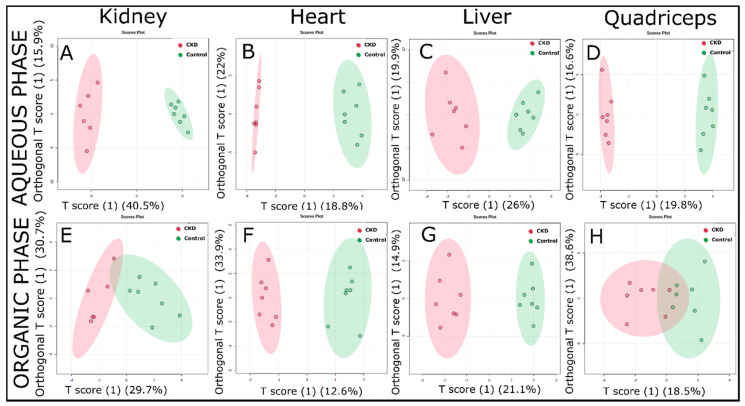
Orthogonal projections to latent structures discriminant analysis (OPLS-DA) score plots obtained from the ^1^H-NMR spectra of the extracted tissue samples (using the targeted profiling approach) from the control and adenine-induced CKD groups. (**A**–**D**) The OPLS-DA score plots from the aqueous phase samples for the kidney, heart, liver, and quadriceps, respectively. (**E**–**H**) The OPLS-DA score plots from the organic phase samples (*n* = 7/group).

**Figure 5 metabolites-11-00045-f005:**
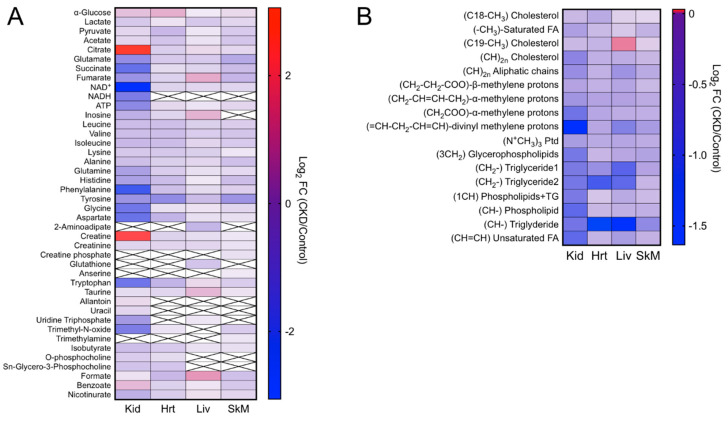
Heatmap representing the log_2_-transformed fold changes (CKD/control) of the metabolites measured in the (**A**) aqueous phase and (**B**) organic phase. Boxes with an “x” inside indicate that the specified metabolite was not detected in the respective tissue (*n* = 7/group). FC: fold change, Kid: kidney, Hrt: heart, Liv: liver, and SkM: skeletal muscle.

**Figure 6 metabolites-11-00045-f006:**
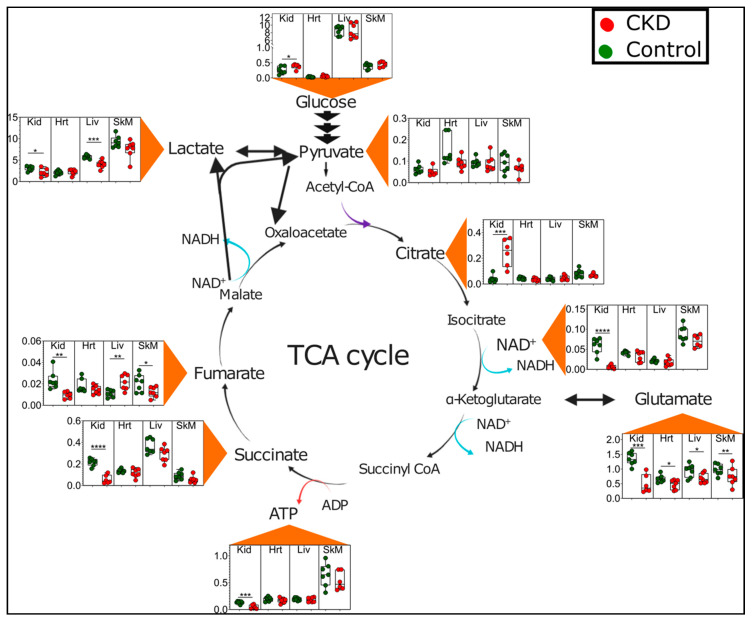
Energetic and tricarboxylic acid (TCA) cycle metabolite disturbances in CKD. Graphical depiction of the TCA cycle with quantified metabolites presented as Box and Whisker plots (showing 95% confidence intervals) in control (green dots) and CKD mice (red dots). The *y*-axis values represent metabolite concentrations in units of millimolars. A two-tailed unpaired *t*-test was performed to determine the statistical significance, with * *p* < 0.05, ** *p* < 0.01, *** *p* < 0.001, and **** *p* < 0.0001. Abbreviations: Kid = kidney, Hrt = heart, Liv = liver, and SkM = skeletal muscle.

**Figure 7 metabolites-11-00045-f007:**
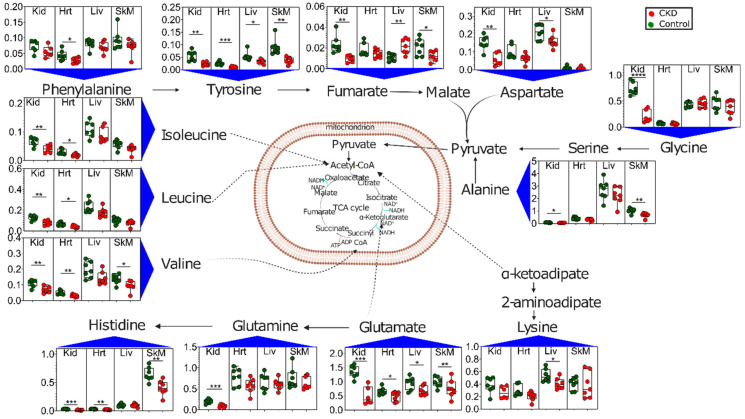
Altered amino acid metabolism in CKD. Graphical depiction of amino acid catabolism with quantified metabolites presented as Box and Whisker plots (showing 95% confidence intervals) in control (green dots) and CKD mice (red dots). The *y*-axis values represent metabolite concentrations in units of millimolars. A two-tailed unpaired *t*-test was performed to determine the statistical significance, with * *p* < 0.05, ** *p* < 0.01, *** *p* < 0.001, and **** *p* < 0.0001 (*n* = 7/group/tissue). Abbreviations: Kid = kidney, Hrt = heart, Liv = liver, and SkM = skeletal muscle.

**Figure 8 metabolites-11-00045-f008:**
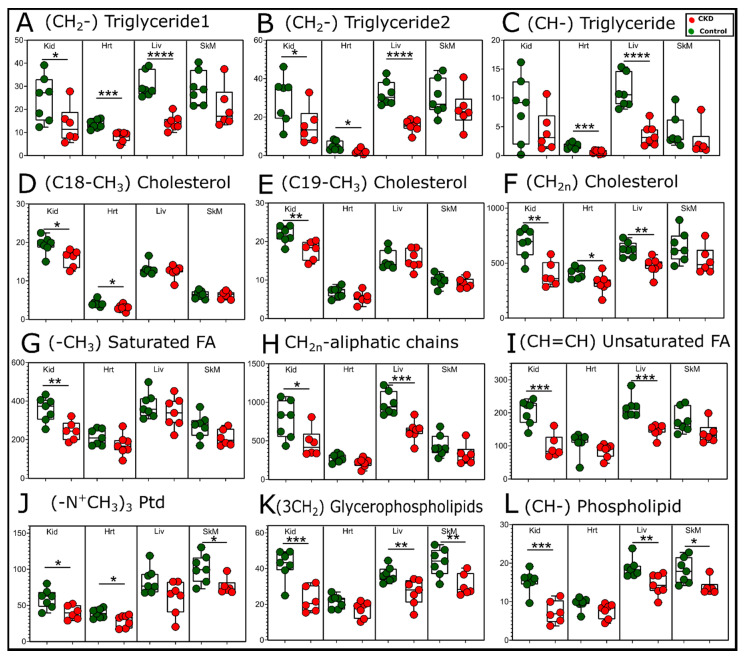
Changes in the lipid compounds with CKD. Quantification of the relative peak areas of selected lipid compounds obtained from organic phase samples presented as Box and Whisker plots (showing 95% confidence intervals). A two-tailed unpaired Student’s *t*-test was performed to determine the statistical significance, with * *p* < 0.05, ** *p* < 0.01, *** *p* < 0.001, and **** *p* < 0.0001 (*n* = 7/group/tissue). Abbreviations: Kid = kidney, Hrt = heart, Liv = liver, and SkM = skeletal muscle.

## Data Availability

All data generated and/or analyzed for this study can be found in Metabolomics Workbench (Project ID: ST001624 to ST001631) and in Appendix A.

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
