# Peer review of "Tissue-Specific ^1^H-NMR Metabolomic Profiling in Mice with Adenine-Induced Chronic Kidney Disease"

_metabolites, 2021, doi:10.3390/metabo11010045_

Round 1

Reviewer 1 Report

This study considered the onset of CKD in mice and the development of a de facto metabolic phenotype for the condition based on induction through to 24 weeks.

Specific comments and included below, whilst review conclusions are summarised at the end of this document.

  • Minor English editing required throughout (line 66 “moreover, disease severity was quite severeevidenced” – “severe” should be modified to “severely”; line 67 “were not used in for…”).
  • Please refer to CKD mice, as opposed to CKD (line 63).
  • In reference to the spectra and Figure 2, please always include the operating frequency at which the measurements were taken. Please also include a table linking the signal ordinal number to identity as opposed to the legend.

  • Finally, a section on developments of the techniques described and future work should be included. This could include a broader view of applying metabolomics in translational medicine (see publications by Roy Goodacre et al.), as well as a wider pool of analytical techniques.

Indeed, CKD is a substantial health burden in domestic animals, particularly cats and current tests are biochemical in nature, with confirmation only through veterinary pathology and MRI assessments, therefore applications of such studies to comparative physiology assessments could prove remarkable for the domestic animal health field.

Author Response

This study considered the onset of CKD in mice and the development of a de facto metabolic phenotype for the condition based on induction through to 24 weeks.

Specific comments and included below, whilst review conclusions are summarised at the end of this document.

Minor English editing required throughout (line 66 “moreover, disease severity was quite severeevidenced” – “severe” should be modified to “severely”; line 67 “were not used in for…”).

We have changed the noted sentence to the following: “Moreover, the clinical disease stage was quite severe evidenced by the spontaneous death of three CKD mice which occurred within two weeks prior to sacrifice”

Please refer to CKD mice, as opposed to CKD (line 63). This correction has been made.

In reference to the spectra and Figure 2, please always include the operating frequency at which the measurements were taken. Please also include a table linking the signal ordinal number to identity as opposed to the legend

We have added the following to Figure 2 legend as suggested: “Proton spectra were collected for both sets of samples (aqueous and organic phases) using a Bruker Avance Neo 14.1 T [600 MHz] (Bruker BioSpin Corporation, Billerica, MA) equipped with a 1.7 mm TCl CryoProbe.”

We appreciated the reviewer’s suggestion to include metabolite identities in table format.  We attempted this approach, but because of the size necessary, adding the table substantially reduces the size of the spectra which becomes difficult for readers to see individual peaks.  Therefore, we prefer to keep metabolite identities in the figure legend to improve the readability of the figure.

Finally, a section on developments of the techniques described and future work should be included. This could include a broader view of applying metabolomics in translational medicine (see publications by Roy Goodacre et al.), as well as a wider pool of analytical techniques

We have added discussion regarding the broader use of metabolomics, future work, and development that can may have strong impact on translational medicine as suggested. This can be found in the last paragraph of the discussion.

Indeed, CKD is a substantial health burden in domestic animals, particularly cats and current tests are biochemical in nature, with confirmation only through veterinary pathology and MRI assessments, therefore applications of such studies to comparative physiology assessments could prove remarkable for the domestic animal health field.

We agree with the reviewer that CKD is a substantial health burden to both humans and domestic animals.  Our study may highlight the potential for more broad discovery and translational medicine to improve the metabolic and health outcomes in the CKD condition.

Reviewer 2 Report

The adenine model is an accurate CKD model that can be constructed easily and showed an important role in the analysis of uremic toxins. The authors' analysis provide important insights into the study of uremic toxins. However, the adenine model has some inconsistencies as a CKD model, so it is necessary to consider.

1) Page9, line 241-244, and Page14, line 476-477.

There are various theories about lipid homeostasis, but the authors' reference (Reference 1) show that triacylglycerols and unsaturated complex lipids was increased in advanced CKD. This is different from the authors' discussion. I recommend that authors add consideration of adaptive mechanism with each stage of CKD.

2) Discussion

The adenine model may differ to human CKD. In particular, the liver is not inflamed which findings can be strongly affected by metabolism (Reference 2), and therefore the model cannot be translated from animals to humans. Therefore, authors need to consider what kind of effect inflammatory response elsewhere than in the kidney has on the amino acids.

References:

  1. Afshinnia, F., et al., Impaired beta-Oxidation and Altered Complex Lipid Fatty Acid Partitioning with Advancing CKD. Journal of the American Society of Nephrology, 2018. 29(1): p. 295-306.
  2. Hamano H. et al. The uremic toxin indoxyl sulfate interferes with iron metabolism by regulating hepcidin in chronic kidney disease,
    Nephrol Dial Transplant. 2018 Apr 1;33(4):586-597.

Author Response

The adenine model is an accurate CKD model that can be constructed easily and showed an important role in the analysis of uremic toxins. The authors' analysis provide important insights into the study of uremic toxins. However, the adenine model has some inconsistencies as a CKD model, so it is necessary to consider.

We agree with the reviewer that the adenine model of CKD has its limitations, as do the various surgical and nephrotoxic models employed in preclinical research. Based on our labs work, we find the adenine model produces greater uremia compared to surgical (5/6 nephrectomy) models which also carry greater mortality rates. We have added some discussion of these issues in the limitations section of the manuscript.

1) Page9, line 241-244, and Page14, line 476-477.

There are various theories about lipid homeostasis, but the authors' reference (Reference 1) show that triacylglycerols and unsaturated complex lipids was increased in advanced CKD. This is different from the authors' discussion. I recommend that authors add consideration of adaptive mechanism with each stage of CKD.

We agree with the reviewer that metabolic changes in CKD are most likely highly dependent on the clinical stage of the disease. We have added some discussion to note this adaptive/clinical stage dependence of the lipidomic changes. Additionally, we would like to the note that the reference raised by the review performed lipid analyses in human plasma, whereas the results presented herein were from tissues.  Plasma lipid profiles are not likely to be the same as organs.

2) Discussion

The adenine model may differ to human CKD. In particular, the liver is not inflamed which findings can be strongly affected by metabolism (Reference 2), and therefore the model cannot be translated from animals to humans. Therefore, authors need to consider what kind of effect inflammatory response elsewhere than in the kidney has on the amino acids.

As discussed above, we acknowledge the limitations of all preclinical CKD models, including the adenine model used in the work. None of the preclinical models are perfect matches to the human condition and disease etiology, however there are clearly some aspects of the model that can inform us of human biology. We have added to the limitation section to further discuss these aspects to inform the readers.

References:

  1. Afshinnia, F., et al., Impaired beta-Oxidation and Altered Complex Lipid Fatty Acid Partitioning with Advancing CKD. Journal of the American Society of Nephrology, 2018. 29(1): p. 295-306.
  2. Hamano H. et al. The uremic toxin indoxyl sulfate interferes with iron metabolism by regulating hepcidin in chronic kidney disease, 
    Nephrol Dial Transplant. 2018 Apr 1;33(4):586-597.

Reviewer 3 Report

This paper by Khattri et al. indicated 1H NMR metabolomic analyses of the heart, liver, kidney, and skeletal muscles obtained from mice with and without CKD. The paper highlighted a metabolomics fingerprint of multiple tissues from mice with and without severe CKD induced by chronic adenine feeding. The paper provides evidence for its conclusion and would be an important read to the metabolomics community. However, the paper is missing valuable information to make physiological conclusions. The corroboration of changes in metabolic pools with physiological data and discussion of their biomedical importance with relevant literature would advance the understanding to influence thinking in the field. It would be important to revise the manuscript with the following comments to make it an interesting read:

Comments

Originality and significance: There is literature available on 1H NMR metabolomic analyses on various organs. The introduction and discussion should be explicitly written to emphasize what extra information this paper provides to advance the thinking in the field.

Data & methodology: How the data binned prior to multivariate analysis using Metaboanalyst?

Statistics: It would be important to report detailed methodology and the normalization method used for multivariate analysis using Metaboanalyst.

Conclusions: The paper indicates alterations in the various metabolic pool but did not include any data to support physiology. It would be essential and useful to include data to support the physiology and corroborate the findings to assess the organ-specific change in metabolic pools e.g. Oxygen consumption etc. The authors should also discuss with relevant literature.

The information on metabolic heterogeneity in various tissues is missing. Specifically, the compartmentalization in kidneys would be discussed.

Clarity and context: The abstract, introduction and discussion should expand the physiological/clinical significance and limitations of the proposed study.

Language: It is advised to proofread the manuscript. Minor language corrections and spell check would be required.

References: Use of reference seems appropriate. However, the discussion section can further be improved by citing more relevant literature.

Figures: Correct the typo in the caption of Figure 7 which is indicated as Figure 8. Axis in Figures 3-4,6-8 is not readable. Also, the axis labeling is missing in Figures 6,7, and 8.

Author Response

This paper by Khattri et al. indicated 1H NMR metabolomic analyses of the heart, liver, kidney, and skeletal muscles obtained from mice with and without CKD. The paper highlighted a metabolomics fingerprint of multiple tissues from mice with and without severe CKD induced by chronic adenine feeding. The paper provides evidence for its conclusion and would be an important read to the metabolomics community. However, the paper is missing valuable information to make physiological conclusions. The corroboration of changes in metabolic pools with physiological data and discussion of their biomedical importance with relevant literature would advance the understanding to influence thinking in the field. It would be important to revise the manuscript with the following comments to make it an interesting read:

Comments

Originality and significance: There is literature available on 1H NMR metabolomic analyses on various organs. The introduction and discussion should be explicitly written to emphasize what extra information this paper provides to advance the thinking in the field.

We acknowledge that other studies have examined metabolomics in CKD, however most of the studies performed analyses at earlier time points – typically 4-6 weeks post CKD induction. We have revised the introduction to make this more clear.

Data & methodology: How the data binned prior to multivariate analysis using Metaboanalyst?

We did not bin data prior to multivariate analyses. We used targeted metabolomics approaches here since there were slight differences in the mass of tissue samples used to acquire 1D 1H NMR. For this, we extracted peak integration areas for several selected metabolites from the pre-processed spectra (using MestReNova 14.1.2-25024 software). Peak areas were correct by the wet weight of the tissue specimen and this normalized peak area was utilized as a raw data for Metaboanalyst.

Statistics: It would be important to report detailed methodology and the normalization method used for multivariate analysis using Metaboanalyst.

Wet weight corrected integrated peak areas for several selected metabolites were utilized as a raw data for Metaboanalyst analysis. In Metaboanalyst analysis, missing value estimation was skipped since our data didn’t contained any missing values. Interquantile range (IQR) filtering was performed in order to get rid of any peaks that did not represent biological metabolites (i.e. NMR noise). Data variability was corrected using probability quotient normalization, followed by pareto-scaling on false discovery rate (FDR) corrected data. Evaluation of PLS-DA and OPLS-DA models were done using Q2-test. This information has been added to the methods of the manuscript.

Conclusions: The paper indicates alterations in the various metabolic pool but did not include any data to support physiology. It would be essential and useful to include data to support the physiology and corroborate the findings to assess the organ-specific change in metabolic pools e.g. Oxygen consumption etc. The authors should also discuss with relevant literature.

We thank the reviewer for this suggestion. However, the aim of this paper was to perform 1H-NMR based metabolomic analyses in organs with relatively high energetic demand in mice with severe CKD. The goal of the study is to understand the global metabolite changes that can help inform us of the potential metabolic interventions that we can develop to improve the condition. Therefore, detailed physiological assessment of each of the organs was not possible.

The information on metabolic heterogeneity in various tissues is missing. Specifically, the compartmentalization in kidneys would be discussed.

This is an excellent point raised by the reviewer. Indeed, each tissue examined herein has a unique structure and contains numerous cell types with distinct functions.  As the technology continues to improve, single-cell analyses may become possible and would facilitate more specific characterization of these tissues. We have included the following in the discussion: “Metabolites were extracted from flash frozen tissue specimens which contain multiple cell types. For example, the kidney has a very distinct structure with many cell types that perform vastly different functions within the tissue [46]. Therefore, the metabolite changes presented herein cannot be directly attributed solely to a single cell type/population within the respective tissues.”

Clarity and context: The abstract, introduction and discussion should expand the physiological/clinical significance and limitations of the proposed study.

We acknowledge the limitations of the metabolomics analyses performed herein, however this was the primary goal of the paper. We prefer to not speculate about the potential physiological/clinical significance of the changes observed in CKD mice.  Inferring physiological significance from these types of analyses without direct experiment evidence demonstrating causality is not appropriate and we feel this is misleading to the readers. Nonetheless, we have revised the discussion to acknowledge the limitation of metabolomics and indicate that future studies are needed to establish causal links.

Language: It is advised to proofread the manuscript. Minor language corrections and spell check would be required. We have revised the manuscript thoroughly to correct these issues.

References: Use of reference seems appropriate. However, the discussion section can further be improved by citing more relevant literature.

We have adjusted the discussion to include some additional relevant papers as suggested.

Figures: Correct the typo in the caption of Figure 7 which is indicated as Figure 8. Axis in Figures 3-4,6-8 is not readable. Also, the axis labeling is missing in Figures 6,7, and 8.

We have corrected this typographical erroer. Y-axis labeling is indicated in the figure legends of all figures as the concentration in millimolar. We prefer to keep this presentation because including the word “millimolar” on every graph is repetitive and distracting due to the number of graph panels in each figure.

Round 2

Reviewer 3 Report

The current version of the manuscript and author's response to previous comments are acceptable.